# Senescence Modulation: An Applied Science Review of Strategies in Anti-Aging, Regenerative Aesthetics, and Oncology Therapy

**DOI:** 10.3390/cimb47120989

**Published:** 2025-11-27

**Authors:** Steven Januar Kusmanto

**Affiliations:** Department of Health Profession, Università San Raffaele Roma, Via di Val Cannuta, 247, 00166 Roma, Italy; stevenjanuarkusmanto@gmail.com

**Keywords:** senescence, anti-aging medicine, aesthetic medicine, oncology, senotherapeutics, senolytics, senomorphics, cancer therapy, targeted therapy

## Abstract

Cellular senescence is an irreversible cell cycle arrest, triggered by stressors like telomere shortening, DNA damage, and oncogenic signaling. These cells, often referred to as ‘zombie cells’ because they cease dividing yet resist apoptosis, drive the Senescence-Associated Secretory Phenotype (SASP), releasing pro-inflammatory cytokines, chemokines, growth factors, and matrix-remodeling enzymes. While senescence is a protective mechanism against malignant proliferation, its persistence in tissues contributes to aging and age-related diseases (inflammaging). Recognizing this dual role forms the basis for developing therapies that bridge anti-aging, regenerative medicine, and oncology, as precise molecular regulatory mechanisms remain incompletely understood. This review interrelates these disciplines, focusing on targeted interventions against senescent cells (SnCs). These interventions include senolytics (agents that selectively eliminate SnCs) and senomorphics (agents that suppress the SASP), offering translational insights from anti-aging/aesthetic applications into integrated treatment models. The framework addresses cancer therapeutics via immunologic modalities such as monoclonal antibodies (mAbs) and CAR T-cell therapy, alongside nucleic acid-based therapeutics (mRNA and siRNA), and is used in combination with broad-spectrum therapeutics. The novelty lies in synthesizing these disparate fields, unified by cellular senescence as a central mechanistic target. Ultimately, the goal is to identify targets that induce tumor regression, mitigate age-related vulnerabilities, promote tissue homeostasis and regeneration, and improve quality of life and overall survival.

## 1. Introduction: Bridging Anti-Aging, Regeneration, and Oncology Through Senescence

Interest in senescence modulation has recently been brought to light as its role in aging, tissue repair, and cancer biology becomes clearer. Cellular senescence provides a common mechanistic foundation linking anti-aging medicine, regenerative aesthetics, and oncology [1]. This review synthesizes the literature to clarify these intersections and identify translational opportunities.

The central challenge in modern oncology today is the dual burden of achieving effective tumor control while simultaneously managing the severe collateral damage inflicted by life-saving treatments. These treatments inflict profound psychosocial hardship through collateral aesthetic damage—such as alopecia, scarring, and premature aging [2,3]—leading to significant psychological distress, social isolation, and an “attack on the self” [4]. The success of reactive aesthetic procedures in restoring self-image and quality of life (QoL), and in enhancing treatment adherence, validates the critical need to integrate appearance-focused care into standard oncology protocols [3,5,6]. The novelty of the proposed framework lies in shifting this essential yet reactive clinical intervention to a proactive molecular level through the principles of cellular senescence (SnC) modulation. This strategy presents a powerful future contribution to molecular medicine by guiding the development of senotherapeutics (senolytics and senomorphics) engineered for a dual therapeutic objective: simultaneous tumor cell elimination and cellular regeneration, directly combating treatment-induced aging and aesthetic damage. Specifically, this framework mandates future clinical development research to focus on identifying robust biomarkers that can simultaneously track the clearance of pro-tumorigenic SnCs in the tumor microenvironment (TME) and pathogenic SnCs in healthy aging tissues. This comprehensive strategy represents the ultimate effort in future cancer care, ensuring that treatment is optimized to improve disease outcome while actively promoting patient well-being, improving confidence, and restoring human dignity. Rather than solely focusing on senescence as a target for symptom management, this review constructs a novel applied science framework that rationalizes and organizes diverse senolytic and senomorphic strategies, positioning them as a unifying translational mechanism across clinical disciplines. This synthesis moves beyond the established biology to focus on the strategic application of senescent cell (SnC) modulation, which is crucial for maximizing the clinical efficacy of traditional therapies. The unique value and forward-looking path of this framework reside in demonstrating how the SnC mechanism, when precisely modulated, achieves two distinct and highly beneficial clinical outcomes: driving desired tissue change (via elimination, e.g., senolytics for anti-aging and regenerative aesthetics) or preventing detrimental changes (via inhibition, e.g., senomorphics/senostatics for oncology) (Figure 1).

Human health and lifespan are closely linked to the aging process, which involves a gradual decline in physiological function and increased vulnerability to diseases, including cancer. Recent developments in senescence research have identified compelling therapeutic targets, ranging from the establishment of integrated treatment models for both regenerative and aesthetic treatments to anti-neoplastic interventions. This review synthesizes the mechanistic underpinnings that interrelate these disciplines, demonstrating how targeted interventions against senescent cells (SnC) provide valuable translational insights for developing integrated treatment models in regenerative medicine and oncology. The novelty of this work lies in its systematic synthesis of perspectives from these seemingly unrelated fields, unified by cellular senescence as a central mechanistic treatment target. This comprehensive analysis, based on an integrated synthesis of the literature, including over 50 key recent publications, is structured to provide a detailed overview of senescence modulation as a unified therapeutic approach. Ultimately, the goal is to identify therapeutic targets that induce tumor regression, mitigate age-related vulnerabilities, and promote tissue homeostasis and regeneration, thereby improving patient-reported outcomes and enhancing overall survival.

## 2. The Fundamental Role of Senescence: A Shared Biological Mechanism

Cellular senescence represents a core biological response to diverse stressors—including DNA damage, telomere attrition, and oncogenic signaling—that collectively drive cells into an irreversible state of arrest of the cell cycle and the establishment of the SnC state (Figure 2) [7,8]. Persistent SnCs, characterized by altered metabolism and apoptosis resistance, become pathogenic through chronic secretion of SASP factors. Crucially, SnCs transform the cellular microenvironment by secreting a complex array of pro-inflammatory cytokines, growth factors, and proteases, collectively known as the Senescence-Associated Secretory Phenotype (SASP). While senescence initially acts as a tumor-suppressive mechanism, the chronic presence of SnCs and their SASP-mediated inflammation drives tissue dysfunction, chronic inflammation (“inflammaging”), and age-related pathology. Senescence subtypes—replicative, oncogene-induced, therapy-induced, and mitochondria-induced—illustrate the diversity of triggers and contexts shaping cellular outcomes [9,10,11,12,13].

The case for cellular senescence as a causal regulator of disease is supported by targeted intervention studies that go beyond correlation to establish mechanistic targetability. This causality is seen across different indications, establishing the SASP as the common pathogenic denominator. In degeneration (aesthetics/aging), the chronic SASP drives an inflammatory feedback loop, which is a necessary cause for age-related tissue remodeling, fibrosis, and impaired regeneration. Recent research confirms that clearing SnC in tissues causally reverses age-related pathology, rather than simply treating symptoms [14]. The removal of uPAR-positive SnCs in preclinical models led to a causal, long-lasting improvement in metabolic health and physical performance, showing that SnCs are the cause of pathology, not just an incidental marker. Likewise, in oncology, the TME depends on the SASP as a proven causal regulator that actively secretes factors (e.g., IL-6, chemokines) that causally promote angiogenesis, metastasis, and, importantly, the immunosuppressive state that directly causes resistance to immune checkpoint inhibitors (ICIs) [15,16]. Therefore, targeting SnCs may offer a strategy to break this pro-tumorigenic and resistance-promoting feedback loop.

## 3. Senescence Modulation in Dermatology, Anti-Aging and Regenerative Aesthetics: A Different Approach to Rejuvenation

In dermatologic and aesthetic medicine, targeting senescence underlies new rejuvenation strategies. Studies have identified key senescence-related genes in dermatologic conditions such as psoriasis [17]. Biologic agents, such as adalimumab (a TNF-α inhibitor), secukinumab (an IL-17A inhibitor), and ustekinumab (an IL-12/23 inhibitor), are effective in targeting specific cytokines. By neutralizing these pro-inflammatory cytokines, these monoclonal antibodies (mAbs) disrupt the inflammatory cascade that drives the characteristic skin lesions of psoriasis, leading to clinical remission and improved patient quality of life [18]. The mechanism of action is linked to their anti-inflammatory properties, which can indirectly modulate the chronic, low-grade inflammation associated with SASP [19]. Also, targeting SnCs holds significant promise in anti-aging and regenerative aesthetics. The accumulation of these cells in the skin contributes to visible signs of aging, including wrinkles, loss of elasticity, and impaired wound healing [20]. Beyond cosmetic improvements, these targets also appear promising in chronic wound care by clearing SnCs that impede tissue repair and regeneration, offering significant advancements in regenerative medicine for dermatological applications [21].

The field of senotherapeutics encompasses a diverse range of strategies, which are broadly classified as senolytics (aimed at eliminating SnCs) and senomorphics (aimed at suppressing the detrimental effects of the SASP). These interventions have emerged as central to anti-aging and regenerative strategies because they target the cellular drivers of inflammaging [22]. For instance, some interventions in preclinical or early clinical research demonstrate the efficacy of these approaches in improving skin health, promoting collagen production, and enhancing tissue regeneration [23,24,25]. The potential to reverse age-related aesthetic changes by directly addressing cellular senescence, which offers a novel and biologically grounded approach compared to traditional cosmetic procedures [25]. Conventional approaches to addressing visible signs of aging, such as skin wrinkling and hair loss, often involve interventions like laser therapies [26,27], dermal fillers [28], and polynucleotide (PN) or polydeoxyribonucleotide (PDRN) injections, which aim to stimulate collagen production and tissue regeneration [29,30]. More advanced strategies include the use of mesenchymal stem cells (MSCs) [31] and various peptides, such as Pep 14, to promote cellular repair and revitalization [32]. While these methods offer variable levels of aesthetic improvement, targeting cellular senescence directly with senolytics or senomorphics presents a novel and biologically grounded approach that could potentially enhance the longevity and efficacy of these traditional procedures by addressing the root cause of age-related tissue dysfunction, offering a promise for more profound and sustained rejuvenation [33].

## 4. Senescence Modulation in Oncology: A Multi-Level Therapeutic Target

Cellular senescence may play a complex, dualistic role in the TME [34], as the definition and function of SnCs vary significantly depending on the tissue type, cellular origin, and specific stressors [35]. The primary therapeutic target is the SnC itself, which must be eliminated or neutralized to prevent tumor recurrence [36]. The key to this strategy lies in recognizing that SnCs and many established tumor cells share overlapping Senescence-Associated Pro-Survival Pathways (SAPS). This shared vulnerability provides a direct translational bridge between oncology and senotherapeutics [36]. For instance, both SnCs and many cancer cells rely on the anti-apoptotic B-cell lymphoma-2 (BCL-2) family of proteins (BCL-2, BCL-xL, and MCL-1) for their survival. This means that many pro-apoptotic cancer therapies developed for malignancies (e.g., BCL-2 inhibitors) often possess inherent senolytic activity, making them immediately relevant for clearing pathogenic SnCs in aging tissues [35,36,37].

Beyond shared survival pathways, senescence also acts as a key effector mechanism in conventional anticancer therapies [38]. For instance, chemotherapy and radiation induce Therapy-Induced Senescence (TIS) in cancer cells, initially acting as a barrier against tumor progression. However, the accumulation of these Therapy-Induced Senescent (TISnt) cells is a significant obstacle to long-term efficacy. This is due to the sustained secretion of the SASP, which remodels the TME to promote inflammation, angiogenesis, and, critically, therapy resistance and cancer recurrence. Furthermore, this SASP-mediated inflammation also drives accelerated aesthetic decline in surrounding healthy tissues, linking the mechanism of recurrence with the mechanism of collateral damage [37]; hence, treatment combination becomes necessary. The combination of traditional pro-senescence therapy (chemotherapy) with senolytics (to clear TISnt cells) [38] or senomorphics (to suppress the detrimental SASP) is a primary strategy currently undergoing rigorous preclinical and clinical investigation to enhance treatment efficacy and reduce adverse effects [34,39].

Emerging modalities, including monoclonal antibodies, antibody-drug conjugates, and CAR-T cell therapy, increasingly intersect with senescence biology. SnCs can create an immunosuppressive TME that dampens the anti-tumor immune response, an effect that immunotherapeutic agents aim to overcome [40]. The efficacy of immune checkpoint inhibitors (ICIs), such as pembrolizumab and nivolumab (which target Programmed Death-1 [PD-1]), is directly linked to overcoming the TME’s immunosuppressive state [41]. Senescence-induced immunosuppression, often driven by components of the SASP, can lead to ICI resistance [42]. This suggests a link between senescence modulation and ICI response. Similarly, the interplay between senescence pathways and the effectiveness of monoclonal antibody (mAb) therapy is emerging as a critical area of investigation [16]. These connections, including a potential link between senescence and cancer immunoediting, are crucial for a comprehensive understanding of the TME. For instance, antibody-drug conjugates (ADCs) serve as a highly targeted delivery system for potent cytotoxic drugs to SASP components that express specific cell surface markers [42]. Neutralizing antibodies also inhibit the release or activity of pro-tumorigenic SASP factors, such as inflammatory cytokines [43]. The use of mAbs, such as adalimumab, primarily used for inflammatory bowel disease and certain autoimmune diseases, and siltuximab, an interleukin-6 (IL-6) inhibitor employed in the treatment of Castleman’s disease, illustrates the principle of modulating inflammatory pathways exacerbated by SnCs [42,44]. This concept may extend to older agents, such as rituximab and regorafenib, whose known clinical effects can also be viewed through the lens of senescence modulation [45,46]. Lastly, chimeric antigen receptor (CAR) T-cells, which have revolutionized the treatment of hematologic malignancies, including lymphomas, leukemias, and myeloma [47], are being engineered to target SnCs specifically. Emerging research indicates that CAR-T cells can be engineered to specifically target SnCs, with targets such as FAP-alpha (Fibroblast Activation Protein-alpha) and uPAR (urokinase plasminogen activator receptor) showing potential to not only enhance the anti-tumor immune response but also mitigate the pro-tumorigenic effects of the SASP within the TME [15,48].

Currently, the two most researched senolytics are the combination of dasatinib, a tyrosine kinase inhibitor, with quercetin, a naturally occurring flavonoid (D + Q) [49], and fisetin, another flavonoid found in various fruits and vegetables [50], both of which are undergoing multiple clinical trials focused on age-related diseases [43].

Lastly, RNA-based therapies (mRNA, siRNA, miRNA) can modulate senescence pathways with unprecedented specificity, representing a frontier in both oncology and regenerative medicine. The prospects of utilizing mRNA, small interfering RNA (siRNA), and microRNA (miRNA)-based therapies to directly target senescence pathways in cancer cells or the surrounding TME represent a significant area for future exploration [51]. The prospect of combining advanced immunotherapeutic modalities with nucleic acid-based therapies to directly modulate senescence pathways in cancer cells or the surrounding TME represents a significant area for future exploration, aiming to improve patient prognosis and overall survival [52].

## 5. Targeting Senescence: Therapeutic Agents and Approaches

### 5.1. Mechanistic Classification of Senolytics (Targeted Elimination)

Senolytics operate by targeting SASP, such as the Bcl-2 family members (e.g., BCL-xL, Bcl-2), which are upregulated in SnCs. Navitoclax (ABT263), for example, is a Bcl-2/BCL-xL inhibitor that induces apoptosis specifically in SnCs [53]. Similarly, the combination of dasatinib (a tyrosine kinase inhibitor targeting Src, PDGFR, etc.) and Quercetin (a pan-inhibitor targeting PI3K/AKT, HSP90) exploits the unique dependence of specific SnC phenotypes on these pathways to induce selective cell death [49]. Ultimately, the objective of targeting senescence is to induce tumor regression, mitigate age-associated vulnerabilities, and promote tissue homeostasis and regeneration, thereby enhancing overall survival and improving patient outcomes [34].

### 5.2. Mechanistic Classification of Senomorphics (SASP Suppression/Inhibition)

On the other hand, senomorphic therapy aims to neutralize the pathogenic effects of SnCs by suppressing them without directly causing their cell death [54]. This is achieved by targeting key signaling hubs that regulate the production of pro-inflammatory factors (IL-6, IL-8). Agents such as metformin suppress the SASP by activating the AMPK/mTOR pathway, thereby reducing the translation of SASP components [55]. Dampening oxidative stress and inflammation can reduce the SASP and the underlying chronic inflammation, as evidenced by its therapeutic applications in type 2 diabetes mellitus and cardiovascular diseases [56]. Ruxolitinib, an inhibitor of Janus kinases (JAKs), is an effective senomorphics because it directly blocks the downstream signaling of pro-inflammatory cytokines like IL-6, thereby creating an autocrine and paracrine feedback loop that propagates the SASP. Its therapeutic efficacy in myelofibrosis and polycythemia vera, conditions marked by excessive inflammation and cytokine dysregulation, underscores its ability to modulate these pathways [57]. Rapamycin, similarly, influences cellular aging and inflammation by directly inhibiting the mechanistic target of the mTOR pathway [58], as described earlier [56]. By suppressing mTOR signaling, rapamycin promotes cellular repair mechanisms, such as autophagy, which clears damaged cellular components, and can reduce the production of SASP factors. This modulation of mTOR can lead to decreased cellular senescence and inflammation, potentially contributing to an extended health span and offering therapeutic avenues for age-related conditions, as suggested by preclinical studies [59]. Its applications in transplant immunosuppression and certain cancers highlight its capacity to modulate fundamental cellular processes [60].

Ongoing research initiatives focus on identifying more specific and safer senolytics and senomorphics, developing biomarkers for monitoring the burden of SnCs, and designing clinical trials to validate these strategies (Figure 3) [61,62]. The potential for personalized medicine, which involves tailoring senescence-modulating therapies to an individual’s unique SnC profile and disease context, represents a significant area for future exploration [61].

## 6. Discussion

This synthesis positions senescent cell modulation as a unifying therapeutic paradigm anchored in its ability to generate two distinct, powerful outcomes: driving desired tissue change through targeted elimination (senolytics in anti-aging/regenerative aesthetics) and preventing detrimental proliferation through inhibition (senomorphics/senostatics in oncology. Although SASP modulation remains valuable, the central focus now lies in identifying and clearing SnCs through context-specific biomarkers and targeted agents.

### 6.1. The Challenge of SnC Heterogeneity and Predictive Biomarker Panels

A critical challenge for the future of senotherapeutics, and a central reason the process is recognized as dynamic and multifaceted, is the heterogeneity and precise definition of the SnC phenotype. The search for a universal SnC marker is confounded by the fact that the constellation of markers varies widely based on the inducing stressor (e.g., replicative, oncogene-induced, or TIS), the specific organ or tissue type, and the cell’s differentiation stage. While markers like SA-β-gal activity, upregulation of p16 and p21, and loss of Lamin B1 are commonly used, their presence is not uniform across all SnCs, particularly in vivo. This variability implies that a senolytic effective against a G1-arrested fibroblast may not eliminate a G2-arrested macrophage, underscoring the necessity for complex, context-specific constellations of markers [63]. The translational roadmap must therefore focus on establishing biomarker panels that combine: (1) Reliable cell-cycle arrest markers (e.g., p16, p21); (2) SASP components (e.g., IL-6, IL-8, CCL2, MMPs) measured in serum or tissue secretomes; and (3) Target-specific markers (e.g., high BCL-2 expression to predict Navitoclax response). This integration is crucial for defining the appropriate therapeutic strategy—whether it is an elimination strategy (senolytics) in aesthetic decline or an inhibition strategy (senomorphics) to stabilize the TME—thereby forming the basis of our applied dual framework. Recent advances, such as those by the NIH SenNet Consortium, are directly addressing this challenge through high-resolution, multi-omic approaches and standardization efforts that move beyond single-marker reliance [64]. Single-cell RNA sequencing (scRNA-seq) and machine learning–driven analysis of senescence gene signatures (e.g., SenMayo, SenSig) have revealed transcriptional diversity and SASP variability across tissues, capturing entire molecular phenotypes missed by traditional low-plex panels. These methods correlate key hallmarks—cell cycle arrest markers (CDKN2A/p16, CDKN1A/p21), persistent DNA damage response (γH2AX foci), nuclear changes (LMNB1 loss, HMGB1 release), and SASP diversity (IL-6, SERPINE1)—with differential susceptibility to senolytic drugs.

### 6.2. Current Consensus

The emerging consensus now supports a multi-marker approach to identify senescent cells in vivo reliably. This includes primary indicators of cell cycle arrest combined with at least two auxiliary markers reflecting SASP activity, metabolic reprogramming, or chromatin remodeling. Such robust identification is essential as the field transitions toward advanced therapies and the implementation of a rigorous Translational Roadmap. A prime example is the development of senolytic CAR-T cells that target uPAR, a senescence-associated surface protein. These living-cell therapies achieve durable and prophylactic clearance of uPAR-positive senescent cells, yielding long-term improvements in metabolic fitness and physical performance in preclinical models—outperforming small-molecule senolytics that require repeated administration [65].

From a translational perspective, combination therapies illustrate the field’s potential. For example, sequential treatment using chemotherapy to induce senescence followed by a senolytic to clear residual SnCs could yield more durable responses. Similarly, pairing immunotherapies with senomorphics may enhance anti-tumor immunity by mitigating SASP-induced immunosuppression. The integration of nucleic acid-based therapeutics offers another avenue for precise modulation of senescence pathways.

The application of nucleic acid-based therapeutics, such as mRNA and siRNA, further enriches this conceptual framework. These technologies offer a high degree of specificity and can be engineered to target the molecular machinery of SnCs or to deliver agents that modulate the SASP. For example, siRNA could be used to silence key SASP genes, while mRNA could be used to express proteins that promote senolytic activity. This level of precision could help overcome the challenges associated with the heterogeneity of SnC populations and their diverse SASP profiles.

Despite the observed convergence in therapeutic strategies—such as the use of classical anti-apoptotic agents (e.g., targeting the BCL-2 family of proteins) as senolytics—it is crucial to recognize that the underlying survival mechanisms in established tumor cells and SnCs are fundamentally distinct. Our framework, however, proposes that the identification of these shared molecular vulnerabilities, even if derived from different pathways, provides a valuable platform for translational research, specifically by addressing the pro-tumorigenic microenvironment created by the SASP. The precise identification and characterization of SnCs across different tissues and disease states require advanced multi-omic approaches.

Moreover, the long-term safety and systemic effects of senotherapeutics in humans need rigorous clinical validation. The potential for these agents to disrupt beneficial, transient senescence—such as that involved in wound healing—must be carefully considered. Nonetheless, the unified conceptual framework presented in this review provides a roadmap for future research.

As the field advances, a multidisciplinary perspective is crucial for bridging the gap between fundamental biological discoveries and their clinical applications. Translating laboratory findings into effective therapeutic regimens will require rigorous preclinical validation and, importantly, thoughtful clinical trial design that measures the framework’s dual goals: oncological survival and QoL/aesthetic improvement. This must be combined with robust biomarker development to monitor both efficacy and safety in fragile cancer survivors. Moreover, the integration of emerging technologies—such as high-throughput screening, single-cell analytics, and advanced computational modeling—holds the promise of accelerating the identification of novel senescence regulators and optimizing patient-specific interventions. Ethical considerations, including the long-term consequences of eliminating or altering the fate of SnCs, must also be addressed to ensure responsible translational progress.

## 7. Conclusions

The convergence of anti-aging medicine, aesthetics, and cancer research around the central theme of senescence modulation highlights the idea that targeting cellular senescence offers a comprehensive approach to addressing age-related decline, enhancing regenerative processes, and improving cancer treatment outcomes. The insights presented in this commentary lay the groundwork for a more integrated and biologically informed approach to addressing some of the most pressing challenges in human health. By shifting focus from SASP suppression to the identification and clearance of SnCs, these disciplines share a mechanistic foundation that enables cross-disciplinary innovation. Key areas for future research lie in a Translational Roadmap that includes the identification of more specific and safer senolytics and senomorphics, the development of robust, context-specific biomarkers to track SnC burden across diverse tissues, and the design of thoughtful clinical trials to validate the efficacy and safety of these dual-strategy approaches in various human diseases and conditions. The potential for personalized medicine approaches that tailor senescence-modulating therapies based on an individual’s specific SnC profile and disease context is a significant point for future exploration.

## Figures and Tables

**Figure 1 cimb-47-00989-f001:**
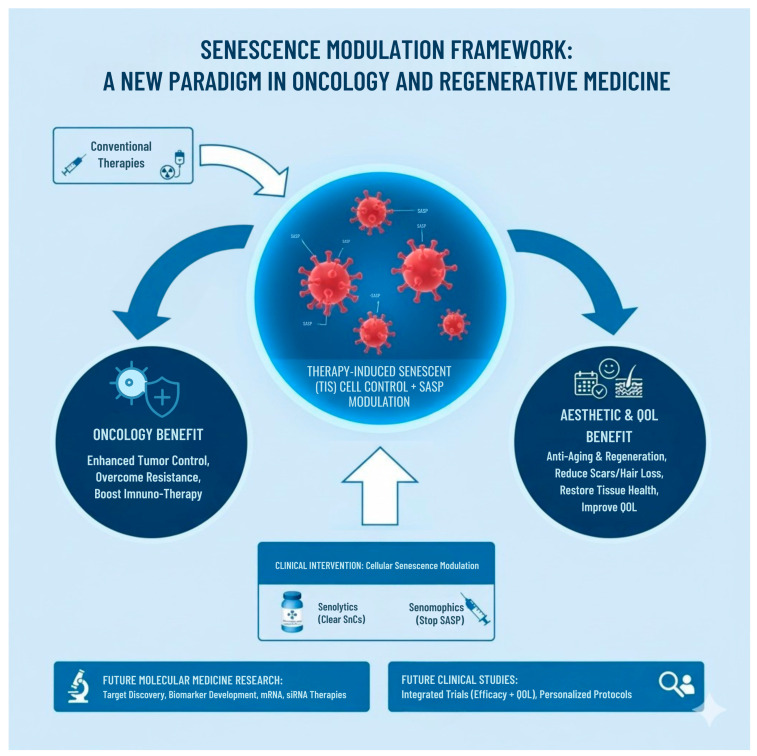
Senescence modulation framework: A new paradigm in oncology and regenerative medicine. mRNA: messenger ribonucleic acid; QOL: quality of life; SASP: Senescence-Associated Secretory Phenotype; siRNA: small interfering ribonucleic acid; SnC: senescent cell; TIS: therapy-induced senescence.

**Figure 2 cimb-47-00989-f002:**
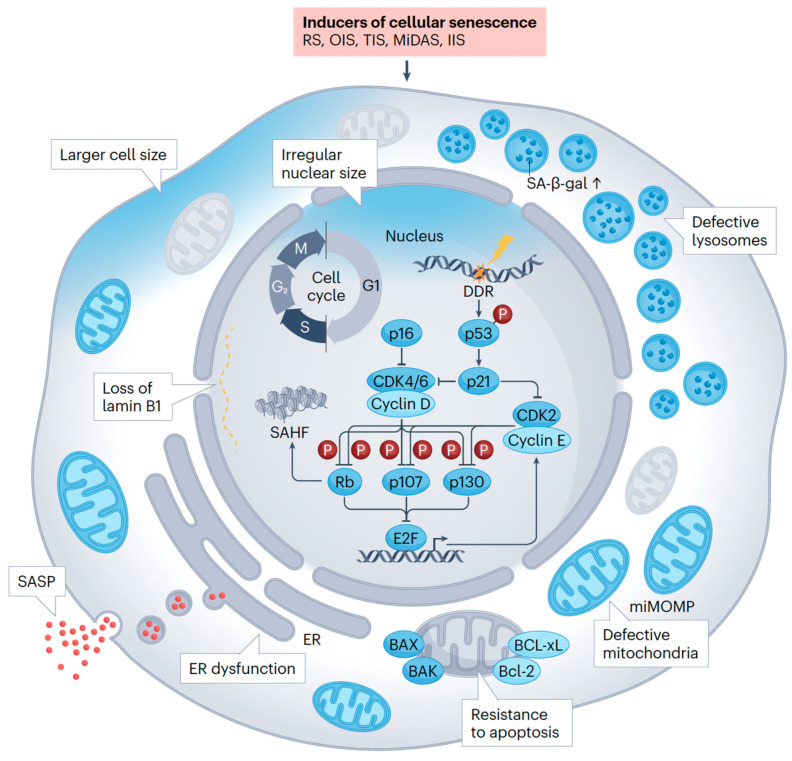
Senescent Cell Characteristics and Classification [7]. Reproduced with permission from Springer Nature. RS: Replicative Senescence; OIS: Oncogene-induced Senescence; TIS: Therapy-induced Senescence; MiDAS: Mitochondria-induced Senescence; IIS: Inflammation-induced Senescence; SA-β-gal: Senescence-Associated β-galactosidase; DDR: DNA Damage Response; CDK: Cyclin-Dependent Kinase; Rb: Retinoblastoma protein; SAHF: Senescence-Associated Heterochromatin Foci; E2F: E2 promoter-binding factor; SASP: Senescence-Associated Secretory Phenotype; ER: Endoplasmic Reticulum; BAX: Bcl-2-associated X protein; BCL-xL: B-cell lymphoma-extra-large; Bcl-2: B-cell lymphoma 2; miMOMP: mitochondrial outer membrane permeabilization.

**Figure 3 cimb-47-00989-f003:**
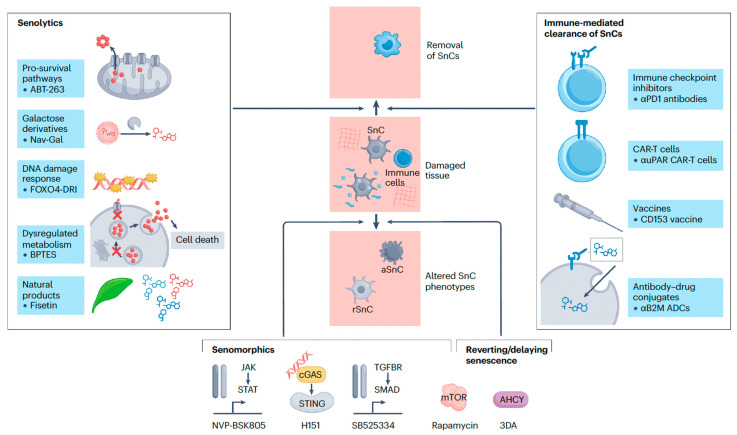
Eliminating Senescent Cells: Therapeutic Strategies for Cancer and Age-Related Disease [62]. Reproduced with permission from Springer Nature. SnC: Senescent Cell; aSnc: Activated Senescent Cell; rSnC: Resident Senescent Cell; αPD1 antibodies: anti-Programmed Death-1 antibodies; CAR-T cells: Chimeric Antigen Receptor T-cells; αuPAR CAR-T cells: urokinase plasminogen activator receptor Chimeric Antigen Receptor T-cells; CD153 vaccine: Cluster of Differentiation 153 vaccine; αB2M ADCs: anti-beta-2-microglobulin Antibody-Drug Conjugates; JAK: Janus Kinase; STAT: Signal Transducer and Activator of Transcription; cGAS: cyclic GMP-AMP synthase; STING: Stimulator of Interferon Genes; TGFBR: Transforming Growth Factor Beta Receptor; SMAD: Small Mothers Against Decapentaplegic Homolog; mTOR: mammalian Target of Rapamycin; AHCY: S-adenosylhomocysteine hydrolase; 3DA: 3-Deazaadenosine; DNA: Deoxyribonucleic Acid.

## Data Availability

No new data were created or analyzed in this study. Data sharing is not applicable to this article.

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
