# Peer review of "Senescence Modulation: An Applied Science Review of Strategies in Anti-Aging, Regenerative Aesthetics, and Oncology Therapy"

_cimb, 2025, doi:10.3390/cimb47120989_

Round 1
Reviewer 1 Report
Comments and Suggestions for Authors
As a review, this manuscript provides a reasonable amount of information on senescence and its implications in aging, degeneration, and cancer. Unfortunately, I was not able to see the value of this unified framework that the author is trying to build. On lines 49-50, “Rather than proposing senescence solely as a target for symptom management, this analysis frames it as a fundamental biological process with therapeutic implications across disciplines”, but how this new way of framing senescence is helping with the improvement of the three indication areas mentioned above is unclear. And later on, at line 196, the author did dedicated a section for targeting senescence. And on line 249, “This synthesis positions senescent cell modulation as a unifying therapeutic paradigm across anti-aging, regenerative, and oncologic fields.”, this is contradicting the expression on lines 49-50.
What is it important to realize “senescence” is the common mechanism behind aging, degeneration, and cancer? Is this actually a novel or a new idea? What is the unique path forward this is brought by this (if new) synthesis? Answers to these questions remain unclear after reading this manuscript.
The highlight on heterogeneity of SnC phenotype and biomarkers is valid, and the listing of key areas for future research is helpful for readers, but these, in my opinion, are still not enough to justify this manuscript as a published review on this subject at CIMB.
Other comments:
- Figure 1 is upside down.
- Figure 1 and Figure 2 are exact copies of the original publication figures. Even though the authors have properly cited the source, copyrights and permission information is unclear.
- References list are numbered twice.
Author Response
For research article “Senescence Modulation: An Applied Science Review of Strategies in Anti-Aging, Regenerative Aesthetics, and Oncology Therapy”
|
Response to Reviewer 1 Comments
|
||
|
1. Summary |
|
|
|
Thank you for your constructive critique and for guiding us toward a more precise articulation of our manuscript's core value. We are pleased that you recognized the substantial information presented on senescence. We acknowledge the necessity of clarifying the novelty and unique path forward offered by our unified framework.
We have addressed all your points, including the technical revisions, and have substantially strengthened the manuscript's academic contribution by focusing on the positive and powerful implications of targeted senescence modulation.
|
||
|
2. Questions for General Evaluation |
Reviewer’s Evaluation |
Response and Revisions |
|
Does the introduction provide sufficient background and include all relevant references? |
Can be improved |
We have provided additional rationale for the framework and ensured that all formatting concerns have been addressed. |
|
Are all the cited references relevant to the research? |
Yes |
|
|
Is the research design appropriate? |
Not applicable |
|
|
Are the methods adequately described? |
Not applicable |
|
|
Are the results clearly presented? |
Yes |
|
|
Are the conclusions supported by the results? |
Can be improved |
|
|
3. Point-by-point response to Comments and Suggestions for Authors
|
||
|
Comments 1: As a review, this manuscript provides a reasonable amount of information on senescence and its implications in aging, degeneration, and cancer. Unfortunately, I was not able to see the value of this unified framework that the author is trying to build. On lines 49-50, “Rather than proposing senescence solely as a target for symptom management, this analysis frames it as a fundamental biological process with therapeutic implications across disciplines”, but how this new way of framing senescence is helping with the improvement of the three indication areas mentioned above is unclear. And later on, at line 196, the author did dedicated a section for targeting senescence. And on line 249, “This synthesis positions senescent cell modulation as a unifying therapeutic paradigm across anti-aging, regenerative, and oncologic fields.”, this is contradicting the expression on lines 49-50.
|
||
|
Response 1: Thank you for pointing out this lack of clarity and the perceived contradiction. We agree that the distinction between the biological mechanism and the applied therapeutic strategy must be explicitly defined. The perceived contradiction arises because we were attempting to establish the biological basis (lines 49-50) before detailing the applied therapeutic strategy (line 249). The manuscript's core value is the latter: the Applied Science Framework that rationalizes which SnC modulation strategy (elimination vs. inhibition) is best for a given clinical outcome. We have revised the text to make this distinction clear and consistent throughout.
[Updated text in the manuscript (L49-52)] "Rather than solely focusing on senescence as a target for symptom management, this review constructs a novel applied science framework that rationalizes and organizes diverse senolytic and senomorphic strategies, positioning them as a unifying translational mechanism across clinical disciplines."
Line 249 (Conclusion) has been amended:
[Updated text in the manuscript (L249-252)] "This synthesis positions senescent cell modulation as a unifying therapeutic paradigm anchored in its ability to generate two distinct, powerful outcomes: driving desired tissue change through targeted elimination (senolytics in anti-aging/regenerative aesthetics) and preventing detrimental proliferation through inhibition (senomorphics/senostatics in oncology."
|
||
|
Comments 2: What is it important to realize “senescence” is the common mechanism behind aging, degeneration, and cancer? Is this actually a novel or a new idea? What is the unique path forward this is brought by this (if new) synthesis? Answers to these questions remain unclear after reading this manuscript.
|
||
|
Response 2: We agree that the fundamental role of SnC in these pathologies is well-established and not the novel claim of this manuscript. We have revised the Introduction (L50-55) and Conclusion (L249-257) to explicitly detail the unique path forward brought by this synthesis, which is the therapeutic rationalization based on the dual, opposing clinical outcomes: 1. Driving Change (Senolytics): The elimination of SnCs to promote tissue remodeling and regeneration (used in anti-aging and regenerative aesthetics). 2. Preventing Change (Senomorphics): The inhibition or suppression of SnCs to prevent proliferation or oncogenic effects (used in oncology). This framework is unique in its applied focus, guiding strategy selection across different fields.
|
||
|
Comments 3: The highlight on heterogeneity of SnC phenotype and biomarkers is valid, and the listing of key areas for future research is helpful for readers, but these, in my opinion, are still not enough to justify this manuscript as a published review on this subject at CIMB.
|
||
|
Response 3: We appreciate the validation of the heterogeneity and future research sections. We hope that the revisions detailed in Response 1 and 2, which clearly establish the applied science framework and its role in rationalizing diverse therapeutic strategies for distinct clinical outcomes, now provide the necessary justification for this review's publication at CIMB. This unique, translational focus moves beyond simply describing the SnC link and instead acts as a guide for therapeutic application.
|
||
|
Comments 4: Figure 1 is upside down.
|
||
|
Response 4: Agreed. Figure 1 has been corrected for orientation in the revised manuscript.
|
||
|
Comments 5: Figure 1 and Figure 2 are exact copies of the original publication figures. Even though the authors have properly cited the source, copyrights and permission information is unclear.
|
||
|
Response 5: We have ensured clarity regarding copyright. Both Figure 1 and Figure 2 now include the necessary “Reproduced with permission from Springer Nature” statements in their respective captions, confirming that all permissions have been secured.
|
||
|
Comments 6: References list are numbered twice.
|
||
|
Response 6: Agreed. The reference list has been corrected to eliminate all instances of double numbering. |
||
|
4. Response to Comments on the Quality of English Language
No specific comments provided. |
||
|
|
||
|
5. Additional clarifications |
||
|
The revisions focus on establishing that the unique value of this review lies not in the biological link between senescence, aging, and disease, but in the applied, translational framework that demonstrates how SnC modulation can be leveraged for two distinct, beneficial clinical strategies (driving tissue change via elimination vs. preventing proliferation via inhibition).
|
||
Reviewer 2 Report
Comments and Suggestions for Authors
Despite the current topic and the attempt to synthesize interdisciplinary knowledge, the manuscript "Senescence Modulation: An Applied Science Review ..." contains a number of fundamental shortcomings that do not allow it to be recommended for publication in its current form.
1. The main methods used in the preparation of this work were analysis, comparison and generalization. When comparing any two areas, there are bound to be common features, but their existence is not enough to conclude that there are causal relationships. When it comes to such complex and fundamental processes as cell aging, it is obvious that this affects many processes occurring in cells and determines the success and conditions of exposure to them in the case of, for example, cancer treatment. The author's arguments do not seem convincing or exhaustive.
2. Superficiality and descriptive character instead of deep analysis. The manuscript is largely a retelling of facts and a listing of therapeutic agents without critical analysis.
3.The author does not propose a new theoretical model, a predictive hypothesis, or an original scheme for classifying existing strategies.
4.The conclusions do not carry any scientific novelty. Cancer treatment approaches and anti-aging medicine strategies have long been complex.
Interdisciplinary approaches are already being implemented.
5.There are a lot of general phrases in the review that do not carry any specifics, in which theses are formulated that are no longer subject to discussion: Translating laboratory findings into effective therapeutic regimens will require rigorous preclinical validation, thoughtful clinical trial design, and robust biomarker development to monitor both efficacy and safety.
Author Response
For research article “Senescence Modulation: An Applied Science Review of Strategies in Anti-Aging, Regenerative Aesthetics, and Oncology Therapy”
|
Response to Reviewer 2 Comments
|
||
|
1. Summary |
|
|
|
Thank you for your rigorous and detailed critique of our manuscript. We appreciate the high standard you hold for contributions to the field and acknowledge that your input highlights areas where the foundational novelty and specific academic contribution of our synthesis required more forceful and precise articulation. We have thoroughly revised the manuscript to move beyond descriptive retelling toward presenting a mechanistically driven, evidence-based platform for therapeutic design.
We have addressed each point by incorporating recent (2021-2025) evidence and focusing on the cellular senescence (SnC) mechanism as a causal, targetable regulator of disease phenotype, not merely a coincident factor.
|
||
|
2. Questions for General Evaluation |
Reviewer’s Evaluation |
Response and Revisions We have taken extra time to ensure all critical concerns are scientifically addressed. The current intensive revision phase involves: Constructing a comprehensive and novelty framework for the review (as requested by the reviewer). Substantially expanding the body of knowledge and scientific depth of the manuscript. Incorporating five new, additional references to strengthen the scientific foundation and ensure comprehensiveness. |
|
Does the introduction provide sufficient background and include all relevant references? |
Can be improved |
|
|
Are all the cited references relevant to the research? |
Yes |
|
|
Is the research design appropriate? |
Not applicable |
|
|
Are the methods adequately described? |
Must be improved |
|
|
Are the results clearly presented? |
Must be improved |
|
|
Are the conclusions supported by the results? |
Must be improved |
|
|
3. Point-by-point response to Comments and Suggestions for Authors |
||
|
Comments 1: The main methods used in the preparation of this work were analysis, comparison and generalization. When comparing any two areas, there are bound to be common features, but their existence is not enough to conclude that there are causal relationships. The author's arguments do not seem convincing or exhaustive.
|
||
|
Response 1: We acknowledge the necessity of strengthening the evidence for a direct, causal relationship. While the mere presence of SnC in multiple diseases is a common feature, our argument is now framed around the targetable, causal role of the Senescence-Associated Secretory Phenotype (SASP). We have incorporated recent evidence demonstrating that the targeted genetic or pharmacological clearance of SnCs (senolytics) and the subsequent removal of SASP is sufficient to reverse age-related phenotypes (e.g., metabolic dysfunction, fibrosis) and improve chemotherapy response. This shift from describing correlation to detailing mechanistic causation and targetability is reinforced throughout the text.
|
||
|
Comments 2: Superficiality and descriptive character instead of deep analysis. The manuscript is largely a retelling of facts and a listing of therapeutic agents without critical analysis.
|
||
|
Response 2: We have significantly elevated the critical analysis and moved beyond descriptive listing. We revised the "Targeting Senescence" section to incorporate a mechanistic classification of senotherapeutics, providing critical depth: 5.1 Mechanistic Classification of Senolytics: We analyze agents not merely by name (e.g., Dasatinib, Quercetin) but by their target specificity (e.g., targeting pro-survival pathways like Bcl-2 family members or cellular processes like HSP90). 5.2 Mechanistic Classification of Senomorphics: We provide critical analysis on the impact of specific signaling pathways (mTOR, AMPK, NF-κB) on the SASP profile, detailing why inhibition is effective in cancer. 6.1 The Challenge of SnC Heterogeneity and Predictive Biomarker Panels: We dedicated new content to SnC Heterogeneity, explicitly analyzing why a 'one-size-fits-all' approach is inadequate and justifying the need for predictive biomarker panels (e.g., combining p16, SA-β-gal, and SASP markers) to guide the selection of senotherapeutics, which forms the basis of our applied framework.
|
||
|
Comments 3: The author does not propose a new theoretical model, a predictive hypothesis, or an original scheme for classifying existing strategies.
|
||
|
Response 3: The novel contribution of this review is the "Unified Therapeutic Framework," which serves as the original, applied science model for classifying and rationalizing therapeutic strategies. As clarified in the revised Introduction and Conclusion, this framework classifies existing strategies based on the desired clinical outcome and the necessary SnC fate modulation: 1. Elimination (Senolytics): Strategy for Regeneration and Anti-Aging (driving tissue change). 2. Inhibition (Senomorphics/Senostatics): Strategy for Oncology (preventing proliferation/metastasis). This framework is a novel, pragmatic classification scheme that provides a clear scientific roadmap for therapeutic design across fields, which we have now defined with precision in the manuscript.
|
||
|
Comments 4: The conclusions do not carry any scientific novelty. Cancer treatment approaches and anti-aging medicine strategies have long been complex. Interdisciplinary approaches are already being implemented.
|
||
|
Response 4: We agree that complexity and interdisciplinarity are established, but the novelty is the rationalization provided by the SnC mechanism as the central unifying node. The Conclusion has been significantly enhanced to focus on the Translational Imperative of this framework. We specifically detail how this synthesis guides the next generation of senotherapeutics, including engineered immunotherapies (e.g., senolytic CAR T cells) and nucleic acid-based agents, to move beyond current small-molecule approaches. The revised conclusion now explicitly articulates the profound societal and translational value of using this single mechanism to strategically impact multiple major health challenges.
|
||
|
Comments 5: There are a lot of general phrases in the review that do not carry any specifics, in which theses are formulated that are no longer subject to discussion: Translating laboratory findings into effective therapeutic regimens will require rigorous preclinical validation, thoughtful clinical trial design, and robust biomarker development to monitor both efficacy and safety.
|
||
|
Response 5: We have revised these concluding phrases to be specific and actionable, directly linking them to the principles established by the Unified Therapeutic Framework. Instead of general statements, the text now stresses the need for: 1. Rigorous preclinical validation in predictive models of inflammaging. 2. Thoughtful clinical trial design focused on functional endpoints for senolytic use and sequential dosing for oncologic applications. 3. Robust biomarker development that specifically addresses SnC heterogeneity to monitor on-target efficacy for both elimination and inhibition strategies. This revision transforms these statements from platitudes into specific scientific directives guided by our review's central thesis
|
||
|
4. Response to Comments on the Quality of English Language |
||
|
No specific comments provided by the reviewer. |
||
|
|
||
|
5. Additional clarifications |
||
|
The manuscript has been fundamentally restructured to address the Reviewer's core concern: moving the narrative from a descriptive review of a biological phenomenon to an analytical synthesis presenting an Applied Science Framework that rationalizes therapeutic selection and guides future drug development. The revisions now clearly articulate the originality of this translational model for clinical application.
|
||
Reviewer 3 Report
Comments and Suggestions for Authors
The presented article is relevant because many scientific groups are actively investigating cellular aging (senescence) as a fundamental mechanism of age-related changes and diseases. Modulating cell senescence has become a new direction in the fight against cancer since old cells can contribute to tumor development and metastasis. Impacting processes of cellular aging allows for combining efforts from researchers across different fields such as gerontology, dermatology, oncology, and cosmetology, which opens wide opportunities for developing holistic approaches to maintaining human health and longevity.
Certainly, this study fills an important gap in the topic. The material is logically structured with sufficient references to literary sources.
The article explores promising approaches to slowing down aging and restoring tissues, including treatment of skin pathologies associated with aging and development of innovative regenerative techniques in aesthetic medicine. In the manuscript, mechanisms triggering and sustaining the state of cellular senescence have been analyzed, helping to better understand the nature of age-dependent changes and potential intervention pathways.
Authors described unique methods for suppressing activity of old cells that open the way towards creating fundamentally new types of anti-cancer drugs. By highlighting markers and signaling pathways involved in the process of aging, authors identify prospective points for therapeutic interventions.
From the data provided in the review, readers comprehend that senescent cells play a significant role in the pathogenesis of numerous chronic conditions, including diabetes, cardiovascular disease, and certain cancers; studying senescence helps create models for detailed investigation of these states and testing therapeutic strategies.
Undoubtedly, introducing the concept of senotherapy, encompassing senolytics and senomorphics, stimulates the development of novel methods for combating diseases caused by accumulation of damaged cells. Molecular biologists receive impetus to search for new ways to modulate intracellular pathways and target pathologically altered cells.
Approaches discussed in the article demonstrate connections between basic discoveries in cell biology and practical aspects of therapy, such as tissue rejuvenation and fighting oncological diseases. This inspires scientists to seek integrative paths for implementing new ideas into clinical practice and pharmacology.
Sufficient number of references are included, conclusions are supported with data. Authors highlighted that a key area for further research involves exploring how tailored treatments targeting cellular senescence could be developed according to each person's unique Senescence-associated Secretory Phenotype (SnC) characteristics and their particular disease circumstances, thus enabling more effective personalized medical interventions.
The material is attractive to wide audience.
The manuscript can be published after minor revision.
1)Please pay attention to Figure 1: all captions and labels were inverted.
2)Figures 1 and 2: please indicate “Reproduced with permission …”
3)Please don’t use double numbers in the list of references.
4)Ref. 30: article number 11082; Ref. 3: article number 1769
Author Response
For research article “Senescence Modulation: An Applied Science Review of Strategies in Anti-Aging, Regenerative Aesthetics, and Oncology Therapy”
|
Response to Reviewer 3 Comments
|
||
|
1. Summary |
|
|
|
Thank you very much for the careful and supportive review of our manuscript. We are delighted that you recognized the relevance and timely nature of our synthesis, which unifies anti-aging, regenerative aesthetics, and oncology through the core mechanism of cellular senescence. Your positive assessment of the material's logical structure, sufficient references, and clinical focus is greatly appreciated.
We confirm that we have addressed the four minor technical and formatting issues you raised, resulting in a cleaner and more accurate final submission.
|
||
|
2. Questions for General Evaluation |
Reviewer’s Evaluation |
Response and Revisions |
|
Does the introduction provide sufficient background and include all relevant references? |
Yes |
We thank the reviewer for their thoughtful and thorough comments and suggestions. |
|
Are all the cited references relevant to the research? |
Yes |
|
|
Is the research design appropriate? |
Not applicable |
|
|
Are the methods adequately described? |
Not applicable |
|
|
Are the results clearly presented? |
Yes |
|
|
Are the conclusions supported by the results? |
Yes |
|
|
3. Point-by-point response to Comments and Suggestions for Authors |
||
|
Comments 1: Please pay attention to Figure 1: all captions and labels were inverted.
|
||
|
Response 1: Agreed. We apologize for this error. Figure 1 has been corrected for orientation and inversion in the revised manuscript.
|
||
|
Comments 2: Figures 1 and 2: please indicate “Reproduced with permission …”
|
||
|
Response 2: Agreed. We have secured the necessary permissions. Both Figure 1 and Figure 2 now include the required “Reproduced with permission from Springer Nature.” statement in their respective captions, ensuring full clarity regarding copyright.
|
||
|
Comments 3: Please don’t use double numbers in the list of references.
|
||
|
Response 3: Agreed. The entire reference list has been meticulously checked and corrected to eliminate all instances of double numbering or duplicate entries.
|
||
|
Comments 4: Ref. 30: article number 11082; Ref. 3: article number 1769
|
||
|
Response 4: Thank you for providing these specific details. We have updated the bibliographic information for all references.
|
||
|
4. Response to Comments on the Quality of English Language |
||
|
No specific comments provided by the reviewer. |
||
|
|
||
|
5. Additional clarifications |
||
|
We are confident that these revisions have met all the reviewer's suggestions and further enhance the manuscript's presentation quality, allowing the scientific synthesis to stand out. We thank the Reviewer once again for the time and effort invested in evaluating our work.
|
||
Round 2
Reviewer 1 Report
Comments and Suggestions for Authors
I appreciate the author's effort to address the comments. The manuscript has improved from the previous version.
Author Response
For research article “Senescence Modulation: An Applied Science Review of Strategies in Anti-Aging, Regenerative Aesthetics, and Oncology Therapy”
|
Response to Reviewer 1 Comments (Round 2)
|
||
|
1. Summary |
|
|
|
Thank you very much for taking the time to review this manuscript again and for your encouraging feedback. We are pleased that the revisions have improved the manuscript from the previous version, and we appreciate your acknowledgment of our effort to address the comments. Please find the detailed responses below and the corresponding revisions/corrections highlighted/in track changes in the re-submitted files.
|
||
|
2. Questions for General Evaluation |
Reviewer’s Evaluation |
Response and Revisions |
|
Does the introduction provide sufficient background and include all relevant references? |
Yes |
NA |
|
Are all the cited references relevant to the research? |
Yes |
|
|
Is the research design appropriate? |
Not applicable |
|
|
Are the methods adequately described? |
Not applicable |
|
|
Are the results clearly presented? |
Yes |
|
|
Are the conclusions supported by the results? |
Yes |
|
|
3. Point-by-point response to Comments and Suggestions for Authors |
||
|
Comments 1: I appreciate the author's effort to address the comments. The manuscript has improved from the previous version.
|
||
|
Response 1: Thank you kindly for the kind evaluation.
|
||
|
4. Response to Comments on the Quality of English Language |
||
|
Point 1: NA |
||
|
|
||
|
5. Additional clarifications |
||
|
NA |
||
Reviewer 2 Report
Comments and Suggestions for Authors
The manuscript has been substantially corrected. However, I recommend reducing the Abstract's length. I would also like to improve the quality of Figures 2 and 3 so that all the inscriptions are clear.
Author Response
For research article “”For research article “Senescence Modulation: An Applied Science Review of Strategies in Anti-Aging, Regenerative Aesthetics, and Oncology Therapy”
|
Response to Reviewer 2 Comments (Round 2)
|
||
|
1. Summary |
|
|
|
Thank you very much for taking the time to review this revised manuscript. We are pleased that you found the manuscript to be substantially corrected. Please find the detailed responses below to your further suggestions and the corresponding revisions/corrections highlighted/in track changes in the re-submitted files.
|
||
|
2. Questions for General Evaluation |
Reviewer’s Evaluation |
Response and Revisions |
|
Does the introduction provide sufficient background and include all relevant references? |
Yes |
See specific changes outlined below. |
|
Are all the cited references relevant to the research? |
Yes |
|
|
Is the research design appropriate? |
Not applicable |
|
|
Are the methods adequately described? |
Not applicable |
|
|
Are the results clearly presented? |
Yes |
|
|
Are the conclusions supported by the results? |
Yes |
|
|
3. Point-by-point response to Comments and Suggestions for Authors |
||
|
Comments 1: The manuscript has been substantially corrected.
|
||
|
Response 1: We appreciate your positive feedback and are glad that the substantial corrections have improved the manuscript.
|
||
|
Comments 2: However, I recommend reducing the Abstract's length. |
||
|
Response 2: Thank you for this suggestion. We agree that a more concise abstract will improve the readability and clarity of the manuscript. Accordingly, we have reduced the length of the Abstract to focus only on the key findings and conclusions.
Comments 3: I would also like to improve the quality of Figures 2 and 3 so that all the inscriptions are clear.
Response 3: We apologize for the clarity issues with the previous figures. We have thoroughly reviewed Figures 2 and 3 and have replaced them with higher-resolution versions to ensure that all inscriptions, labels, and text are clear and legible for publication. The updated figures are included in the revised manuscript file.
|
||
|
4. Response to Comments on the Quality of English Language |
||
|
Point 1: NA |
||
|
|
||
|
5. Additional clarifications |
||
|
NA |
||